# Titratable Pharmacological Regulation of CAR T Cells Using Zinc Finger-Based Transcription Factors

**DOI:** 10.3390/cancers13194741

**Published:** 2021-09-22

**Authors:** Bettina Kotter, Fabian Engert, Winfried Krueger, Andre Roy, Wael Al Rawashdeh, Nicole Cordes, Britta Drees, Brian Webster, Niels Werchau, Dominik Lock, Sandra Dapa, Dina Schneider, Stephan Ludwig, Claudia Rossig, Mario Assenmacher, Joerg Mittelstaet, Andrew D. Kaiser

**Affiliations:** 1Miltenyi Biotec B.V. & Co. KG, Friedrich-Ebert-Straße 68, 51429 Bergisch Gladbach, Germany; bettinak@miltenyi.com (B.K.); fabiane@miltenyi.com (F.E.); waela@miltenyi.com (W.A.R.); nicolec@miltenyi.com (N.C.); brittad@miltenyi.com (B.D.); brianwe@miltenyi.com (B.W.); nielsw@miltenyi.com (N.W.); dominiklo@miltenyi.com (D.L.); sandrad@miltenyi.com (S.D.); mario@miltenyi.com (M.A.); andrewk@miltenyi.com (A.D.K.); 2Lentigen Technology Inc., A Miltenyi Biotec Company, 910 Clopper Road, Suite 200 S, Gaithersburg, MD 20878, USA; Winfried.Krueger@miltenyi.com (W.K.); Andre.Roy@miltenyi.com (A.R.); Dina.Schneider@miltenyi.com (D.S.); 3Interdisciplinary Center for Clinical Research (IZKF) and Institute of Virology (IVM), University Hospital Muenster, WWU, Von-Esmarch-Straße 56, 48149 Muenster, Germany; ludwigs@uni-muenster.de; 4Department of Pediatric Hematology and Oncology, University Children’s Hospital Muenster, Albert-Schweitzer-Campus 1, 48149 Muenster, Germany; rossig@uni-muenster.de

**Keywords:** cellular immunotherapy, chimeric antigen receptor T cells (CAR T cells), transcriptional control

## Abstract

**Simple Summary:**

Chimeric antigen receptor (CAR) T cell therapy can be associated with substantial side effects primarily due to intense immune activation following treatment, or target antigen recognition on off-tumor tissue. Consequently, temporal and tunable control of CAR T cell activity is of major importance for the clinical translation of innovative CAR designs. This work demonstrates the transcriptional regulation of an anti-CD20 CAR in primary T cells using a drug inducible zinc finger-based transcription factor. The switch system enables titratable induction of CAR expression and CAR T cell effector function with the clinically relevant inducer drug tamoxifen and its metabolites both in vitro and in vivo, whereby CAR activity is strictly dependent on the presence of the inducer drug. The results obtained can readily be transferred to other CARs for which an improved control of expression is required.

**Abstract:**

Chimeric antigen receptor (CAR) T cell therapy has emerged as an attractive strategy for cancer immunotherapy. Despite remarkable success for hematological malignancies, excessive activity and poor control of CAR T cells can result in severe adverse events requiring control strategies to improve safety. This work illustrates the feasibility of a zinc finger-based inducible switch system for transcriptional regulation of an anti-CD20 CAR in primary T cells providing small molecule-inducible control over therapeutic functions. We demonstrate time- and dose-dependent induction of anti-CD20 CAR expression and function with metabolites of the clinically-approved drug tamoxifen, and the absence of background CAR activity in the non-induced state. Inducible CAR T cells executed fine-tuned cytolytic activity against target cells both in vitro and in vivo, whereas CAR-related functions were lost upon drug discontinuation. This zinc finger-based transcriptional control system can be extended to other therapeutically important CARs, thus paving the way for safer cellular therapies.

## 1. Introduction

Adoptive transfer of CAR-modified T cells is emerging as a promising treatment modality for a broad range of cancers. Most advanced in current clinical development is the treatment of B cell malignancies including acute and chronic B cell leukemia as well as B cell non-Hodgkin lymphoma with anti-CD19 CAR T cells that have recently been approved by the FDA [1,2,3]. Despite a pooled overall response rate of 71% across patients with B cell malignancies refractory to standard therapies [4], CAR T cell therapy is associated with unique acute and chronic side effects [5]. Cytokine release syndrome and neurotoxicity, the most commonly observed immediate adverse events, result from the excessive activity of CAR T cells upon antigen stimulation [6,7]. In addition, even low expression of the target antigen on normal tissue can result in on-target/off-tumor toxicities. In patients with B cell malignancies, on-target depletion of normal B cells causes B cell aplasia and resultant hypogammaglobulinemia [8,9]. Long-term follow-up studies have reported low remaining IgG levels for up to 4 years correlating with the persistence of the anti-CD19 CAR T cells resulting in a need for immunoglobulin replacement to reduce the susceptibility to infection in many patients [10,11,12]. On-target/off-tumor toxicities of CARs targeting alternative, non-B cell antigens may cause life-threatening toxicities if tissues of vital organs are damaged [13,14].

Different strategies have been developed to improve the safety of CAR T cell therapy [5]. Drug-based suicide switches as well as antibody-mediated depletion mechanisms are under clinical investigation for the elimination of adoptively transferred T cells [15,16,17,18,19]. Furthermore, several molecular approaches to control CAR T cells in a non-apoptotic manner have been evaluated to date [20,21,22,23]. Control of CAR-mediated function at the protein level has been achieved with anti-tag CAR technologies [21,22,24], reversible pharmacological on and off switches interfering with CAR signaling [23], or small molecule gated CARs where CAR function depends on the presence of a small molecule dimerizing the co-stimulatory domain and the CD3ζ chain [20]. Targeting CAR expression rather than CAR signaling, a Tet-on-based transcription regulating circuitry represents a versatile tool for gene specific regulation [25]. However, high background expression in the non-induced state prevents the tight control of transgene expression in primary T cells limiting clinical application of the Tet-on system [26,27].

Alternatively, polydactyl zinc finger proteins assembled of modular zinc finger domains that recognize three contiguous base pairs of the DNA sequence each, can be designed to recognize DNA segments of specific length and sequence [28,29,30,31]. The attachment of an effector domain provides artificial transcription factors that efficiently activate, suppress, or create defined changes to the targeted promoters of both transgenes and endogenous genes [32,33,34]. Control of the transcription factor activity was achieved by fusing zinc finger domains to modified ligand-binding domains of steroid hormone receptors including the estrogen and progesterone receptor [35]. Key for achieving drug-targetable regulation is the unresponsiveness of the ligand-binding domains to the natural steroid ligands. By using a point-mutated, ligand-binding domain of the estrogen receptor in combination with the three finger zinc finger protein N1, Beerli and colleagues reported the efficient induction of reporter constructs with the inducer drug 4-hydroxytamoxifen (4-OHT), a hydroxylated metabolite of the FDA-approved drug tamoxifen, with no response to estrogen [35]. However, to our knowledge, synthetic zinc finger-based systems have not been used in primary T cells to control therapeutically relevant CARs.

Facing the hazardous side effects of CAR T cell therapy there is a high demand for the controlled regulation of transgene expression in adoptively transferred T cells. We employed a zinc finger-based 4-OHT inducible system to fine-tune CAR expression and, thereby, tightly control the cytolytic activity of CAR T cells in vitro and in vivo in a reversible manner. In the present study, we used the anti-CD20 CAR (scFv clone: Leu-16) as a model due to its well-established and published efficacy in tumor eradication [36]. In contrast to existing control strategies, the switch is not integrated into the CAR design and, thus, the functionally validated CAR structure is not disturbed. Using the investigated zinc finger-based system, virtually any CAR under clinical development could be controlled in an on/off switch manner characterized by high flexibility, selectivity, and tight controllability.

## 2. Results

### 2.1. Design of the Inducible Anti-CD20 CAR Construct

For drug controlled transcription of the anti-CD20 CAR, a synthetic transcription factor composed of the N1 zinc finger protein, a modified ligand-binding domain of the estrogen receptor (G525R), and the transcriptional activation domain VP64 was constructed based on sequences reported by Beerli et al. [35] (Figure 1a). The synthetic transcription factor is constitutively expressed under the control of the human PGK promoter and linked via a P2A ribosome skip site to a truncated version of the low-affinity nerve growth factor receptor (∆LNGFR), which serves as a transduction marker. In the presence of 4-OHT, the synthetic transcription factor binds to its respective binding sites and drives the transcription of the anti-CD20 CAR (Figure 1b). The second-generation anti-CD20 CAR is composed of the leader sequence from human granulocyte-macrophage colony-stimulating factor receptor alpha subunit (huGM-CSFR), a single-chain variable fragment targeting CD20, fused to the hinge and transmembrane domain of T cell surface glycoprotein CD8 alpha chain, and the intracellular signaling domains of the co-stimulatory receptor 4-1BB (CD137) followed by the intracytoplasmic region of CD3ζ [36,37]. The inducible element (binding sites—E1bmin—anti-CD20 CAR) was inserted into the vector in 3′ to 5′ direction to avoid read through and background transcription in the absence of the inducer drug. As a positive control, a constitutively expressed anti-CD20 CAR, referred to as direct CAR, was placed under the PGK promoter and linked to ∆LNGFR via the P2A sequence.

### 2.2. Drug Dependent Induction of Anti-CD20 CAR Expression and Induction Dynamics

To study the induction capacity of this system, we transduced isolated primary human T cells with the drug-inducible lentiviral vector and induced the expression of the anti-CD20 CAR by the addition of 100 nM 4-OHT to the cell culture medium. Anti-CD20 CAR expression on the surface of transduced (∆LNGFR^+^) T cells was measured by flow cytometry and revealed the selective induction of anti-CD20 CAR expression within 40 h of exposure to 100 nM 4-OHT. No background expression was detectable in the non-induced state (Figure 1c and Appendix A). Upon induction, inducible CAR T cells expressed significantly higher levels of anti-CD20 CAR, as measured by mean fluorescence intensity (MFI), compared to the conventional constitutively expressed CAR under the control of the PGK promoter (Figure 1d). Anti-CD20 CAR positive T cells could be detected as early as 4 h post-induction (Figure 1e). The frequency of anti-CD20 CAR expressing T cells as well as the CAR surface density (represented as MFI) increased over time and plateaued 8 h after the addition of 100 nM 4-OHT to the culture. Background levels of transgene expression were detected 10 days after discontinuation of 4OHT treatment for inducible anti-CD20 CAR constructs (Figure 1f and Appendix A). The persistence of the anti-CD20 CAR is likely due to surface stability of the protein. Therefore, destabilized GFP was used as a transgene to further characterize the system. Fast on and off switching by the zinc finger-based transcription factor was shown for destabilized GFP (Figure 1g,h). While expression rates were comparable (Figure 1g), decay rates were significantly faster for T cells induced to express a destabilized GFP reaching background expression levels 9 h after 4-OHT removal (Figure 1h and Appendix A). These data point to tight transcriptional regulation with fast off-rates, and the absence of (neo-)transcription after inducer drug removal with transgene persistence depending mainly on the stability and half-life of the protein product.

### 2.3. Inducible CAR T Cells Show Efficient Cytotoxic Activity in the Presence of 4-OHT, While CAR-Related Functions Are Lost after 4-OHT Discontinuation In Vitro

To assess the cytolytic activity of inducible CAR T cells, we performed co-cultures with CD20^+^ target cells in the presence or absence of 100 nM 4-OHT. GFP^+^ 526-Mel cells were genetically modified to express high levels of the CD20 target antigen (Appendix A). Upon induction, inducible anti-CD20 CAR T cells efficiently eradicated GFP^+^ CD20^+^ 526-Mel and achieved the same level of specific lysis as constitutively expressed control CAR T cells, whereas we did not detect target cell lysis in parallel cultures without 4-OHT (Figure 2a). In addition, cytokine secretion by inducible CAR T cells upon antigen engagement was strictly dependent on the presence of 4-OHT with cytokine levels below background in the non-induced state. The induction of CAR T cells resulted in a 400-fold increase in IFN-γ levels and a 70,000-fold increase in IL-2 levels, respectively. While IL-2 levels secreted by inducible CAR T cells in co-cultures with GFP^+^ CD20^+^ 526-Mel were significantly higher compared to the conventional constitutively expressed CAR, IFN-γ levels were comparable (Figure 2b,c). Maximum specific lysis of GFP^+^ CD20^+^ 526-Mel was obtained already at a concentration of 10 nM 4-OHT (EC_50_ = 3.6 nM), thus matching the physiological concentration of 4-OHT detected in breast cancer patients treated with tamoxifen [38]. Endoxifen, a less potent but more prevalent metabolite of tamoxifen, induced cytolytic activity at a higher EC_50_ value of = 28.2 nM (Figure 2d). In vitro, inducible CAR T cells eradicated GFP^+^ CD20^+^ 526-Mel about as efficiently as conventional constitutively expressed CAR T cells even at low effector-to-target cell (E–T) ratios (Figure 2e) demonstrating serial killing capacity and induced killing even if 4-OHT was added at later time points to the co-culture (Figure 2f). Next, we investigated whether activated, inducible CAR T cells lose CAR-related functions after 4-OHT discontinuation. Following efficient lysis of CD20^+^ tumor cells (round 1), T cells were washed and re-co-cultured with freshly plated tumor cells. Although 4-OHT was removed, inducible CAR T cells continued to eradicate GFP^+^ CD20^+^ 526-Mel (Figure 2g and Appendix A). We reasoned that tumor cell lysis was mediated by the remaining anti-CD20 CAR molecules rather than the synthesis of new proteins. Accordingly, we performed a sequential stimulation study. In the third serial co-culture, cytolytic activity was lost in the absence of 4-OHT. While the conventional constitutively expressed anti-CD20 CAR did not eradicate tumor cells in the third round, inducible CAR T cells controlled tumor outgrowth in the presence of 4-OHT (Figure 2g).

### 2.4. Titration of Inducer Dose and Reduction of Response Elements Enables Fine-Tuning of Inducible Anti-CD20 CAR Activity

We titrated the 4-OHT concentration and reduced the number of response elements for the synthetic transcription factor to investigate whether we can fine-tune anti-CD20 CAR expression and, thus, precisely control the cytolytic activity of inducible CAR T cells. Titration experiments revealed that with an increasing drug dose and an increasing number of binding sites per construct, the frequency of T cells expressing the anti-CD20 CAR over detection limit can be maximized (Figure 3a). However, surface expression (represented as MFI) did not reach saturation within the titrated range (Figure 3b). Importantly, for all constructs, CAR expression in T cells was absolutely dependent on the presence of the inducer drug, and strict transcriptional control was maintained after polyclonal activation (Appendix A). CAR-mediated activity was efficiently triggered in T cells incorporating constructs with two or more binding sites and was promoted with increasing concentrations of 4-OHT. Different combinations of the two modulatory entities (supplementation of 4-OHT and number of binding sites) resulted in distinct levels of IFN-γ and IL-2 as well as killing potential in co-culture with GFP^+^ CD20^+^ 526-Mel cells and allowed for a controlled, fine-tuned on switch (Figure 3c–e). Similar results were obtained for co-cultures with the Burkitt lymphoma cell line Raji (Figure 3f–h). Inducible anti-CD20 CAR T cells efficiently eradicated GFP^+^ Raji cells in a dose-dependent manner and achieved the same level of specific lysis as conventional constitutively expressed CAR T cells, whereas we did not detect target cell lysis in parallel cultures without 4-OHT (Figure 3h and Appendix A). IFN-γ and IL-2 secretion were strictly dependent on the presence of the inducer drug (in this case 4-OHT) and cytokine levels could be modulated by the concentration of 4-OHT as well as by the number of binding sites (Figure 3f,g and Appendix A).

### 2.5. Inducible CAR T Cells Efficiently Eradicate Disseminated Lymphoma In Vivo

As tight expression control is a key feature of drug-inducible CAR T cells, an in vivo study was designed to detect any potential residual CAR activity in the absence of the inducer drug, as well as to analyze therapeutic activity of the CD20 CAR T cells in its presence. Therefore, we applied a progressive model of Burkitt lymphoma (Raji^ffLuc^ cells) in immunodeficient NOD.Cg-*Prkdc^scid^ IL-2rg^tm1Wjl^*/SzJ (NSG) mice. Mice received 3 × 10^6^ transduced (∆LNGFR^+^) T cells in a total of 1.1 × 10^7^ T cells, or 1.1 × 10^7^ control T cells intravenously (i.v.) on day 0. Tamoxifen was administered by intraperitoneal (i.p.) injections to the indicated groups starting on day 1. Tumor burden was assessed by bioluminescent imaging (BLI, Figure 4a). Inducible CAR T cells in combination with tamoxifen treatment completely eradicated disseminated lymphoma within 19 days of the T cell injection. Killing kinetics were slightly delayed compared to the conventional constitutively expressed anti-CD20 CAR. In contrast, tumor burden (seen in photon flux in Figure 4c) in the inducible CAR group not receiving tamoxifen progressed similarly to the control groups (untransduced T cells, untreated), demonstrating strict transcriptional control of the inducible CAR cassette. Importantly, tamoxifen-treated mice in the inducible CAR group remained in complete remission throughout the study course of 35 days (Figure 4b,c). Dosing experiments demonstrated that the cytolytic activity of inducible CAR T cells can be controlled in vivo by the tamoxifen dose and the injection frequency (Appendix A). The highest anti-CD20 CAR expression on T cells in the bone marrow was detected in the groups treated with daily injections of 1 mg tamoxifen, while less tamoxifen or an increased administration interval resulted in less effective induction and, thus, reduced tumor control (Appendix A). Active metabolites of tamoxifen reached plasma levels, which could effectively induce ex vivo killing of tumor cells (Appendix A). Plasma cytokine analysis demonstrated a specific tamoxifen-dependent activation of inducible CAR T cells in vivo. Significant levels of IFN-γ and IL-2 were detected in the plasma on day 2 of mice receiving inducible CAR T cells in combination with daily tamoxifen injections compared to control groups, while only background secretion was detected in the non-induced state (Figure 4d,e). Inducible T cells, isolated from the spleen on day 35 after the T cell injection, could be re-induced ex vivo to efficiently express the anti-CD20 CAR by the administration of 100 nM 4-OHT, demonstrating the possibility to control therapy by on and off switching (Figure 4f and Appendix A).

## 3. Discussion

CAR T cell therapy targeting the CD19 antigen has demonstrated significant clinical benefit in hematological malignancies [1,2,3]. However, along with therapy, unwanted side effects have been reported including late effects that can persist throughout the life span of the transferred T cells [5]. Effective toxicity management and prevention requires control over CAR T cell functions after administration into patients, raising the need for a reversible remote control acting in an on/off manner.

In the present study, we demonstrated the use of a tamoxifen-dependent inducible control system based on a previously described transcription factor [35] for the tight control of anti-CD20 CAR expression in primary T cells. Our data illustrate specific induction of anti-CD20 CAR expression in the presence of the inducer drug with no background expression in the non-induced state, differentiating this system from other transcriptional control systems, in particular the Tet-on system for CAR regulation [26,39]. Maximum expression levels for the inducible CAR were up to 7-fold higher compared to the conventional constitutively expressed anti-CD20 CAR (Figure 1d), presumably due to the distinct promoter composition in conjunction with the recruited transcription factors [40]. Despite differences in CAR expression levels between the inducible CAR and the conventional constitutively expressed CAR, tumor cell lysis and IFN-γ secretion in vitro was equally efficient (Figure 2a,b and Appendix A). Previous studies reported an impaired anti-tumor effect of T cells with low CAR expression compared to high CAR levels, especially at low antigen densities [41,42,43]. On the other hand, CAR expression beyond the threshold for effective T cell activation was reported to contribute to a tonic signaling signature and T cell differentiation impairing the in vivo anti-tumor activity [44]. The possible implications of inducible CAR T cells enabling the precise regulation of receptor densities on the T cell surface beyond baseline transcriptional control need to be further evaluated in settings where the target antigen is expressed at lower and/or heterogeneous densities on tumor cells. In particular, with regard to reported differences between in vitro and in vivo studies in terms of optimal CAR expression levels, precise tuning of CAR densities and their adjustment to the patient’s conditions might allow CAR T cells to discriminate between different biological contexts and, thereby, become safe and also effective against tumor cells that share target antigens with normal tissues.

EC_50_ concentrations of 4-OHT and endoxifen required for the specific lysis of tumor cells in vitro are below the physiological concentration of the hydroxylated tamoxifen metabolites that can readily be found in breast cancer patients treated with a daily oral dose of 20 mg tamoxifen [38] (Figure 2d). Endoxifen is present in the blood at higher concentrations than 4-OHT, while 4-OHT displays a higher affinity for the estrogen receptor [38,45]. Side effects of tamoxifen are mainly associated with its demethylated metabolites and related to its hormonal mode of action. These include, among others, hot flushes, fatigue, vaginal dryness, sleep problems, weight gain, mood swings, and depression [46]. Patients treated with tamoxifen over a period of 5 years have been reported to be at higher risk for thromboembolic events and the development of endometrial cancer during the active treatment period [47]. Overall, tamoxifen-related side effects have been well described, are clinically manageable, and might be acceptable in the context of controlling CAR T cell activity, especially if subtherapeutic doses can be used. Furthermore, the direct application of active metabolites requires even lower doses for effective induction [48] and, thereby, could further minimize the risk of associated side effects.

Conventional kill switches are off switches resulting in an irreversible termination of CAR T cell activity. The activation of drug-based suicide switches, including herpes simplex virus thymidine kinase and dimerizing death molecules (Fas, iCas9), results in cell death within 30 min of administration [15,16,17]. For selective antibody-mediated depletion, cells are engineered to express a targetable moiety such as CD20, tEGFR, or CD34 with subsequent apoptosis triggered by the administration of the cognate antibody [18,19]. Besides suboptimal expression levels and the resistance of cells to kill switches limiting the efficacy, the irreversible termination of the CAR-mediated anti-tumor effect might be problematic if the patient requires reactivation [15,16,17].

In contrast, our system allows the management of toxicity by terminating tamoxifen administration, resulting in pausing of the CAR T cell-mediated effects with the possibility of re-inducing CAR expression at later time points. Our study revealed that the re-induction of CAR expression after an on/off phase is feasible, demonstrating complete reversibility of CAR activity (Figure 4f and Appendix A). Thereby, reinfusion of the CAR T cells in case of relapse can be avoided, minimizing patient stress and lowering financial burden. A potential limitation of the tamoxifen-inducible system for the control of anti-CD20 CARs is the slow transition to the off state seen in vitro, as well as the half-life of tamoxifen in vivo. Based on the decay rate determined for destabilized GFP, we hypothesize that the system itself has fast switching kinetics, but it is limited by the stability of the anti-CD20 CAR on the T cell surface (Figure 1e–h and Appendix A).

The relatively long half-life of tamoxifen [49] as well as the lag period until CAR densities reach below the critical threshold required for effector function (Figure 2g), will limit rapid intervention, e.g., for the treatment of life-threatening cytokine release syndrome. However, through combination therapy, e.g., with tocilizumab [50] providing IL-6 receptor blockade, acute adverse events can be controlled temporarily. During this time window, downregulation of CAR expression and associated activities could be realized, enabling the reversible suspension of the therapy. In clinical situations where very rapid intervention is not required, the tamoxifen-regulated system could significantly contribute to the quality of life for patients. The suspension of anti-CD20 CAR T cell function may, for instance, allow the reversal of long-term B cell aplasia and, thus, replace the need for extended, possibly lifelong immunoglobulin substitution. In this regard, the tamoxifen-regulated system would promote the application of B-lineage targeted CAR T cell therapies even in curative settings where immunoglobulin replacement becomes limiting. 

During repeated antigen exposure with alternating on/off states, inducible CAR T cells effectively eradicated CD20+ tumor cells, while the conventional constitutively expressed CAR did not elicit anti-tumor activity in the third round of antigen encounter (Figure 2g). The loss of robust effector function by the conventional constitutively expressed CAR T cells might result from a state of exhaustion due to sustained CAR signaling [51,52]. Defined, alternating on and off phases of CAR expression and, thus, calibration of the activation potential, could reduce chronic signaling and prevent exhaustion [23,53,54]. In this way, titratable CARs could have high therapeutic value far beyond their increased safety.

This proof of concept study demonstrated the use of tamoxifen-regulated zinc finger proteins for the controlled and fine-tuned onset of transgene expression in primary T cells. The activity of inducible anti-CD20 CAR T cells was strictly dependent on the presence of the inducer drug with no background CAR expression, anti-tumor activity, or cytokine secretion in the non-induced state both in vitro and in vivo. The absence of background expression of the 4-OHT-regulated transcriptional control system studied here distinguishes it from the Tet-on system with reported basal expression of anti-CD19 CAR [26,27] in the absence of doxycycline. Moreover, the possibility of regulating the transcriptional output via the number of zinc finger binding sites, adds an additional modality for the adjustment of the transcriptional output beyond drug dosing. The system described here is under further investigation for translation towards clinical use and is intended to become a flexible platform applicable to any CAR T cell therapy beyond B cell malignancies, e.g., in solid tumors, where temporal and tunable control of CAR T cell function will be essential to increase safety. For clinical application, a detailed risk analysis focusing on the potential immunogenicity of the synthetic transcription factor will be performed.

## 4. Material and Methods

### 4.1. Cell Lines

For in vitro experiments, the Burkitt lymphoma cell line Raji as well as the melanoma cell line 526-Mel were transduced with lentiviral vectors encoding a GFP-firefly luciferase cassette. GFP^+^ 526-Mel cells were additionally modified to stably express CD20. For in vivo studies, a mouse-adapted Raji^ffLuc^ cell line was used [37]. All tumor cell lines were cultured in RPMI 1640 (Biowest, Nuaillé, France) supplemented with 2 mM glutamine (Lonza, Basel, Switzerland) and 10% fetal bovine serum (FBS, Biochrome, Berlin, Germany). The human embryonic kidney cell line 293 T was cultured in Dulbecco’s modified Eagle medium (Biowest, Nuaillé, France) supplemented with 10% FBS.

### 4.2. Generation of CAR Constructs and Lentiviral Vector Production

The sequence of the synthetic transcription factor was ordered (ATUM, Newark, CA, USA) and cloned into a second-generation vector system (Miltenyi Biotec, Bergisch Gladbach, Germany) whereby the constitutive and inducible part were incorporated in an inverted orientation. The construction of the inducible anti-CD20 CAR plasmid was based on Beerli et al. [35]. The synthetic transcription factor composed of the N1 zinc finger, the murine estrogen receptor (G525R), and VP64 are constitutively expressed under the regulation of the human PGK promoter. ∆LNGFR serves as transduction marker as well as enrichment tag and is co-expressed with the synthetic transcription factor at equimolar levels from a single RNA transcript via a P2A sequence [55]. The synthetic transcription factor binds to up to five response elements upstream of the E1b minimal promoter and induces the transcription of a 2nd generation anti-CD20 CAR incorporating a leader sequence from huGM-CSFR, an anti-CD20 scFv based on Leu-16, CD8 hinge, and transmembrane domain, as well as the cytoplasmic domains of 4-1BB and CD3ζ [37]. For experiments with focus on induction dynamics, the anti-CD20 CAR cassette was exchanged to a sequence encoding destabilized GFP [56]. The conventional anti-CD20 CAR is constitutively expressed under the control of the human PGK promoter and is based on the same sequence as the inducible anti-CD20 CAR. Lentiviral particles were generated by the transient transfection of HEK 293 T cells using polyethylenimine. LV containing supernatants were concentrated by overnight centrifugation (5380 g, 24 h) and pelleted LVs were resuspended in PBS before storing at −80 °C. LV titers were determined by the transduction of SupT1 and flow cytometric analysis of ∆LNGFR expression.

### 4.3. Generation and Expansion of Inducible CAR T Cells

Buffy coats and non-mobilized leukapheresis were obtained from healthy volunteer donors from the DRK Dortmund and Ulm. Human PBMCs were purified by density gradient centrifugation using Pancoll solution (Pan-Biotech, Aidenbach, Germany). Primary human T cells were isolated from PBMCs by negative bead selection according to manufacturer’s recommendations (Pan T cell isolation kit, Miltenyi Biotec, Bergisch Gladbach, Germany) and activated using MACS GMP T Cell TransAct (Miltenyi Biotec, Bergisch Gladbach, Germany). T cells were cultured in TexMACS GMP medium (Miltenyi Biotec, Bergisch Gladbach, Germany) supplemented with 12.5 ng/mL of recombinant human IL-7 and 12.5 ng/mL of recombinant human IL-15 (Miltenyi Biotec, Bergisch Gladbach, Germany). T cells were transduced using lentiviral vectors 24 h after stimulation using an MOI of 5 to maintain the vector copy number of the drug product below 5. T cells were washed 3 days after stimulation to remove MACS GMP T Cell TransAct and lentiviral particles. T cells were expanded for 13–14 days with the addition of fresh culture medium every 2–3 days, before the experiments were started. Timing of T cell generation, including activation, transduction, and addition of medium was kept consistent for experiments comparing multiple donors and CAR constructs.

### 4.4. Induction of Transgene Expression

T cell numbers were adjusted to 5 × 10^4^ transduced T cells/mL and transgene transcription was induced by the addition of the indicated concentration of (Z)-4-hydroxytamoxifen (Sigma-Aldrich, St. Louis, MO, USA), (Z)-endoxifen (Axon Medchem, Groningen, The Netherlands) or a 1:9 mixture thereof (stock: 1 µM (Z)-4-hydroxytamoxifen, 9 µM (Z)-endoxifen) directly to the culture medium. Transgene expression was analyzed at the indicated time point by flow cytometry.

### 4.5. Flow Cytometry Analysis

Transduction efficiency was determined by flow cytometry staining of T cells using an APC-labelled anti-LNGFR antibody according to manufacturer’s recommendations (10 min at 4–8 °C, Miltenyi Biotec, Bergisch Gladbach, Germany). Anti-CD20 CAR expression was detected using anti-CD20 CAR detection reagent linked to PE (10 min at 4–8 °C, Miltenyi Biotec, Bergisch Gladbach, Germany). To exclude dead cells, propidium iodide was added to the stained cells directly before sample acquisition at the MACSQuant Analyzer. MACSQuantify 2.13.0 or Flow Logic software 7.2.1 was used for data analysis.

### 4.6. Cellular Cytotoxicity Assay

T cells were co-cultured with 1 × 10^4^ target cells (GFP^+^ Raji or GFP^+^ CD20^+^ 526-Mel) at indicated effector-to-target cell (E–T) ratios in the presence of stated concentrations of 4-OHT, endoxifen, or a combination thereof. Prior to each experiment, T cell numbers were adjusted to LNGFR expression implicating equal transduced and total T cell numbers in each well. To assess the cytolytic activity of inducible CAR T cells either the outgrowth of tumor cells was monitored over time using a live-cell imaging device (IncuCyte, Essen BioScience, Ann Arbor, MI, USA) or specific lysis was determined after the quantification of living target cells using the MACS Quant Analyzer 40 h after co-culture set-up. Specific lysis was calculated according to the following formula: % specific lysis = 1 − (cell count [sample]/cell count [target cell]). EC_50_ values were obtained by fitting log-transformed and normalized data using non-linear regression with GraphPad Prism 8. All samples were run in duplicates.

### 4.7. Cytokine Secretion Assay

Transduced as well as untransduced T cells were cultured with targets cells in a 1:1 ratio (1 × 10^4^ cells per well each) in the presence of the indicated concentration of 4-OHT, endoxifen or a combination thereof. After 40 h, the cytokine secretion of T cells was determined in the supernatant of the co-culture using MACSPlex Cytokine 12 Kit, human (Miltenyi Biotec, Bergisch Gladbach, Germany), according to the manufacturer’s protocol.

### 4.8. Xenograft Mouse Model and Bioluminescent Imaging

All animal experiments were approved by the ethical committee on animal care and use in Nordrhein-Westfalen, Germany (approval number: 81-02.04.2018.A326). For tumor establishment 3 × 10^5^ or 4 × 10^5^ mouse-adapted Raji^ffLuc^ cells were injected i.v. into the tail vein of NSG mice (NOD.Cg-*Prkdc^scid^IL2rg^tm1 Wjl^*/SzJ). Tumor growth was monitored using an IVIS Lumina III instrument (Perkin Elmer, Waltham, MA, USA) 6 min after the i.p. injection of 100 μL (30 mg/mL) D-Luciferin (K^+^ Salt) Bioluminescent Substrate (GoldBio, St Louis, MO, USA). Images were analyzed using Living Image, version 4.5.2., software (Perkin Elmer, Waltham, MA, USA) and the bioluminescent signal for each mouse was expressed in photon flux (photons per second). 3 × 10^6^ transduced T cells (in total 1.1 × 10^7^ T cells) were infused i.v. 5 days after tumor engraftment. Non-transduced T cells and an untreated group served as controls. Tamoxifen (Sigma-Aldrich, St. Louis, MO, USA) dispersed in 50 µL peanut oil (Sigma-Aldrich, St. Louis, MO, USA) was administered i.p. every 24 h or 48 h to untreated (=tumor only), untransduced, and inducible CAR w/Tam groups. Direct CAR and inducible CAR w/o groups received oil injections without tamoxifen. Blood samples were collected from the *Vena facialis* on day 2 post T cell injection and processed to generate plasma for the measurement of plasma cytokines via MACSPlex Cytokine 12 Kit, human (Miltenyi Biotec, Bergisch Gladbach, Germany). For ex vivo analysis, mice were sacrificed and femur and tibia of both legs were flushed with RPMI to retrieve bone marrow isolates. Spleens were disrupted manually. Red blood cells were lysed and cells were stained for flow cytometry analysis or expanded ex vivo after restimulation.

### 4.9. Statistical Analysis

Statistical analysis was performed using GraphPad Prism 8 software (GraphPad, San Diego, CA, USA). All data are represented as mean ± s.d. unless otherwise noted. Data were analyzed for statistical differences using tests indicated in the figure legends.

## 5. Conclusions

This study demonstrates the feasibility of CAR expression control in primary T cells using a zinc finger-based transcription factor that is responsive to metabolites of the clinically approved drug tamoxifen. The system is characterized by absence of background expression, and enabled time- and dose-dependent induction of anti-CD20 CAR expression, as well as fine-tuning of cytolytic activity against target cells, both in vitro and in vivo. This system may be readily applied for regulation of other CARs for which an improved control of expression is required.

## Figures and Tables

**Figure 1 cancers-13-04741-f001:**
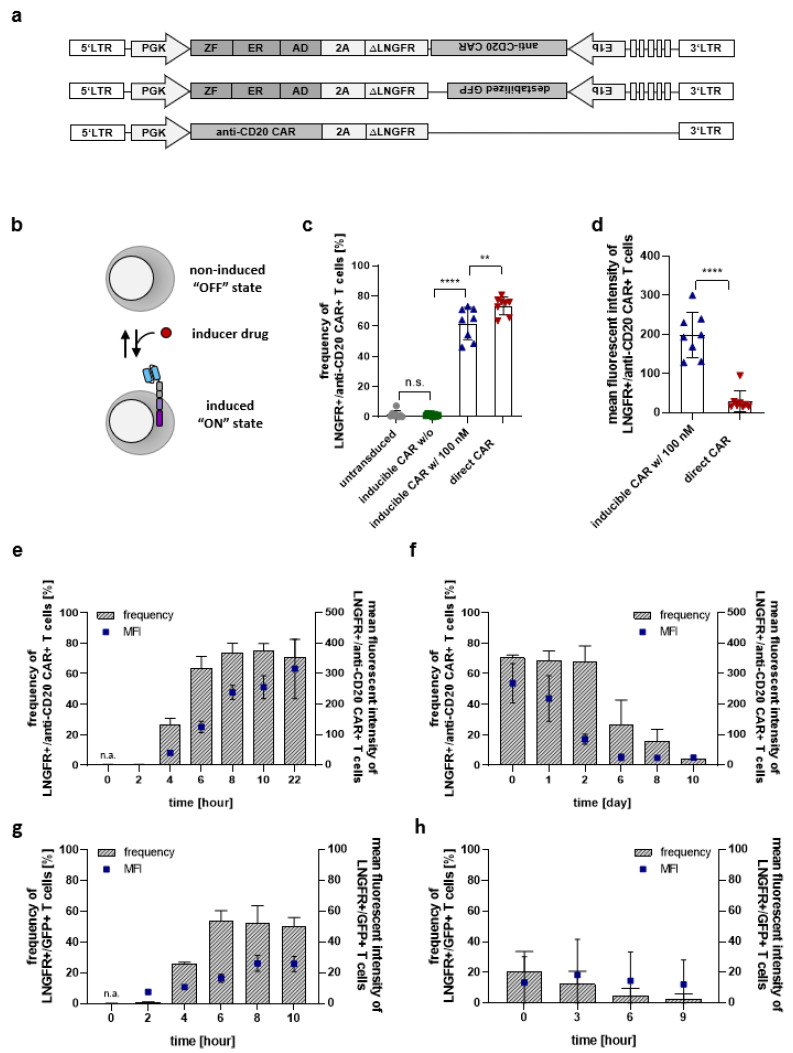
Drug dependent induction of anti-CD20 CAR expression and induction rates. (**a**) Schematic representation of the inducible anti-CD20 CAR, inducible destabilized GFP, and the conventional CAR construct. The constitutively expressed synthetic transcription factor is composed of the N1 zinc finger (ZF), an estrogen receptor (ER), and VP64 (AD). Transcription of the anti-CD20 CAR or destabilized GFP is initiated upon binding to its response elements upstream of the E1b minimal promoter. The conventional anti-CD20 CAR is constitutively expressed under the control of the PGK promoter and represents the positive control. ∆LNGFR is included in both constructs as a transduction marker. (**b**) Schematic illustration of the mode of action. In the presence of the inducer drug (4-OHT, endoxifen) transcription and, thus, the expression of the anti-CD20 CAR in primary T cells are triggered. In the absence of the inducer, T cells do not express the anti-CD20 CAR. (**c**,**d**) Cell surface expression of anti-CD20 CAR amongst ∆LNGFR expressing cells was determined 40 h after the induction with 100 nM 4-OHT by flow cytometry using anti-CD20 CAR detection reagent-PE. Data for frequency (**c**) and MFI (**d**) of anti-CD20 CAR T cells are normalized to the expression of the transduction marker ∆LNGFR. (**e**) Cell surface expression of anti-CD20 CAR was analyzed at indicated time points after the induction with 100 nM 4-OHT by flow cytometry and normalized to the expression of ∆LNGFR. (**f**) Inducible CAR T cells were induced with 100 nM for 40 h and anti-CD20 CAR expression was determined for the indicated time points after 4-OHT discontinuation. (**g**) Expression of destabilized GFP was analyzed at indicated time points in primary T cells after the induction with 100 nM 4-OHT by flow cytometry and normalized to the expression of ∆LNGFR. (**h**) GFP expression in T cells was induced with 100 nM 4-OHT for 40 h and determined for the indicated time points after 4-OHT discontinuation. Graphs represent data from *n* = 8 (**c**,**d**); *n* = 3 (**e**–**h**) different donors. Data shown are mean values ± s.d. with ** *p* < 0.01, **** *p* < 0.0001 by one-way analysis of variance (ANOVA) (**c**), or by unpaired two-tailed t test (**d**). n.s.: not statistically significant.

**Figure 2 cancers-13-04741-f002:**
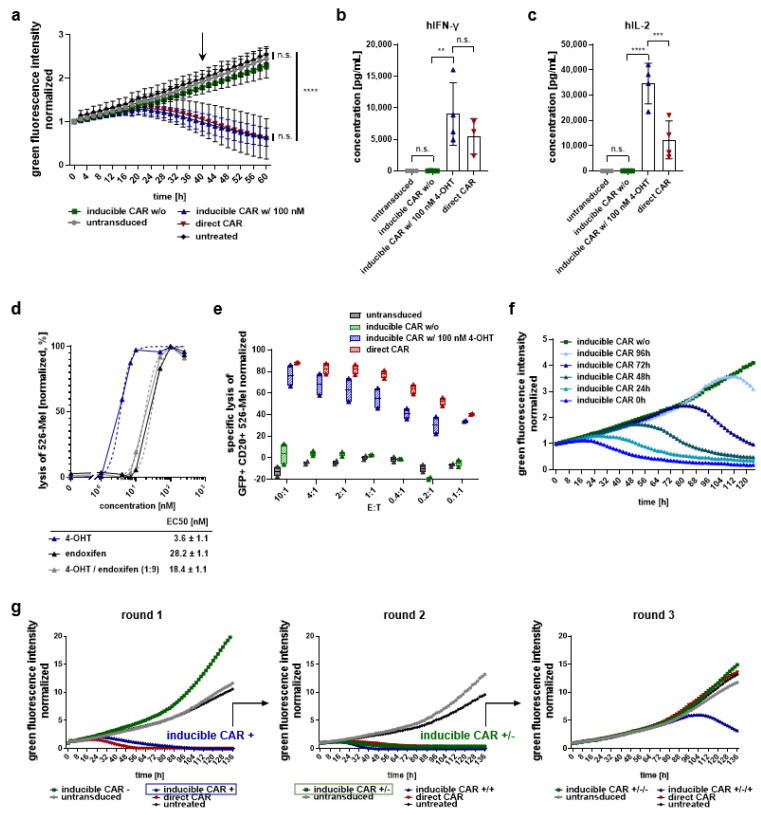
Inducible CAR T cells show efficient cytotoxic activity in the presence of 4-OHT, while CAR-related functions are lost after 4-OHT discontinuation in vitro. (**a**) T cells expressing ∆LNGFR were co-cultured with GFP^+^ CD20+ 526-Mel at an E–T ratio of 1:1 in the presence of 100 nM 4-OHT added at the start of the assay. T cell numbers were adjusted to LNGFR expression implicating equal transduced and total T cell numbers in each well. Growth of GFP^+^ CD20^+^ 526-Mel was monitored in 2 h intervals over a period of 60 h using a live-cell imaging device (IncuCyte). The arrow indicates the time point of cytokine analysis. (**b**,**c**) The concentration of human IFN-γ (**b**) and IL-2 (**c**) was analyzed in the supernatant 40 h after assay initiation by MACSPlex Cytokine 12 Kit. (**d**) Inducible CAR T cells were co-cultured with GFP^+^ CD20^+^ 526-Mel at an E–T ratio of 1:1 in the presence of varying concentrations of 4-OHT and/or endoxifen. Specific lysis of GFP^+^ CD20^+^ 526-Mel was measured 40 h after co-culture setup, log transformed and normalized. Non-linear regression was performed to calculate EC_50_ values. (**e**) Cytolytic activity of T cells against GFP^+^ CD20^+^ 526-Mel at varying E–T ratios was assessed 40 h after co-culture set-up in the presence or absence of 100 nM 4-OHT. (**f**) The inducer drug 4-OHT was added at different time points at a concentration of 100 nM to the co-culture of inducible CAR T cells and GFP^+^ CD20^+^ 526-Mel. (**g**) T cells were co-cultured with GFP^+^ CD20^+^ 526-Mel at an E–T ratio of 1:1 in the presence of 100 nM 4-OHT added at the start of the assay. After each co-culture, T cells were harvested and 4-OHT was removed. T cells were then transferred onto new tumor cells in the absence (-) or presence (+) of 100 nM 4-OHT. Growth of GFP^+^ CD20^+^ 526-Mel was monitored in 2 h intervals and expressed as normalized green fluorescence intensity. In (**f**,**g**) graphs represent pooled results of different donors generated within one experiment. Graphs show data from *n* = 5 (**a**); *n* = 4 (**b**,**c**); *n* = 3 (**d**); *n* = 2 (**e**–**g**) different donors. Data shown are mean values ± s.d. with ** *p* < 0.01, *** *p* < 0.001, **** *p* < 0.0001 by two-way analysis of variance (ANOVA) (**a**), or by one-way ANOVA (**b**,**c**). n.s.: not statistically significant.

**Figure 3 cancers-13-04741-f003:**
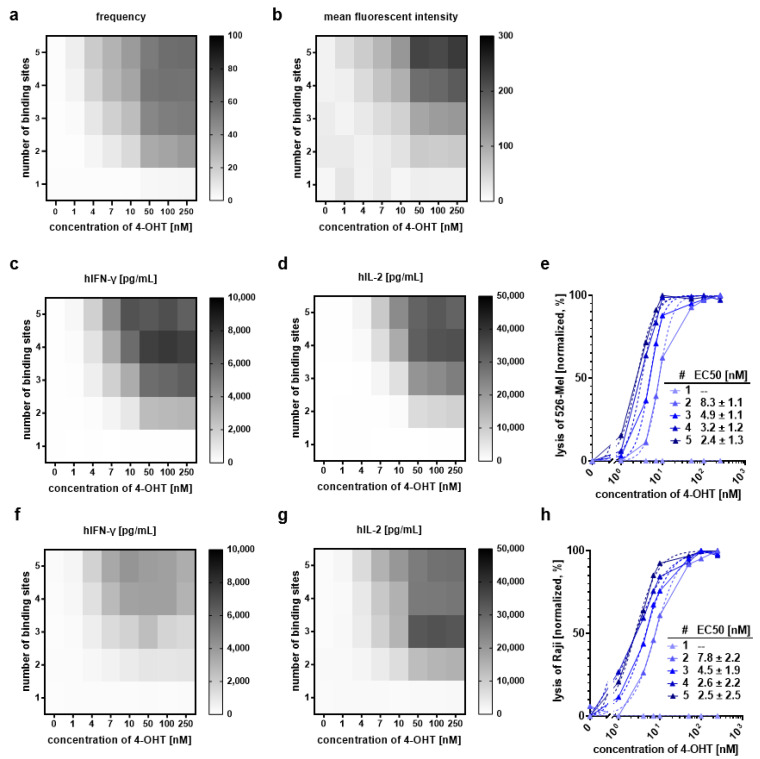
Modulation of anti-CD20 CAR induction by the inducer dose and the number of response elements. (**a**,**b**) Inducible CAR T cells bearing one to five binding sites for the synthetic transcription factor were induced with varying concentrations of 4-OHT. Cell surface expression of anti-CD20 CAR was determined after 40 h by flow cytometry using anti-CD20 CAR detection reagent-PE. Data for frequency (**a**) and MFI (**b**) of anti-CD20 CAR T cells are normalized to the expression of the transduction marker ∆LNGFR. (**c**–**h**) Inducible CAR T cells bearing one to five binding sites for the synthetic transcription factor were co-cultured with GFP^+^ CD20^+^ 526-Mel (upper panel) or GFP^+^ Raji (lower panel) for 40 h at an E–T ratio of 1:1 in the presence of varying concentrations of 4-OHT. (**c**,**d**,**f**,**g**) Human IFN-γ and IL-2 levels were detected in the culture supernatant by MACSPlex Cytokine 12 Kit. (**e**,**h**) Titration curves were determined for the lysis of GFP^+^ CD20^+^ 526-Mel and GFP^+^ Raji. Non-linear regression was performed to calculate EC_50_ values following log transformation and normalization of data. Graphs show data from *n* = 5 (**a**,**b**,**e**,**h**); *n* = 2 (**c**,**d**,**f**,**g**) different donors.

**Figure 4 cancers-13-04741-f004:**
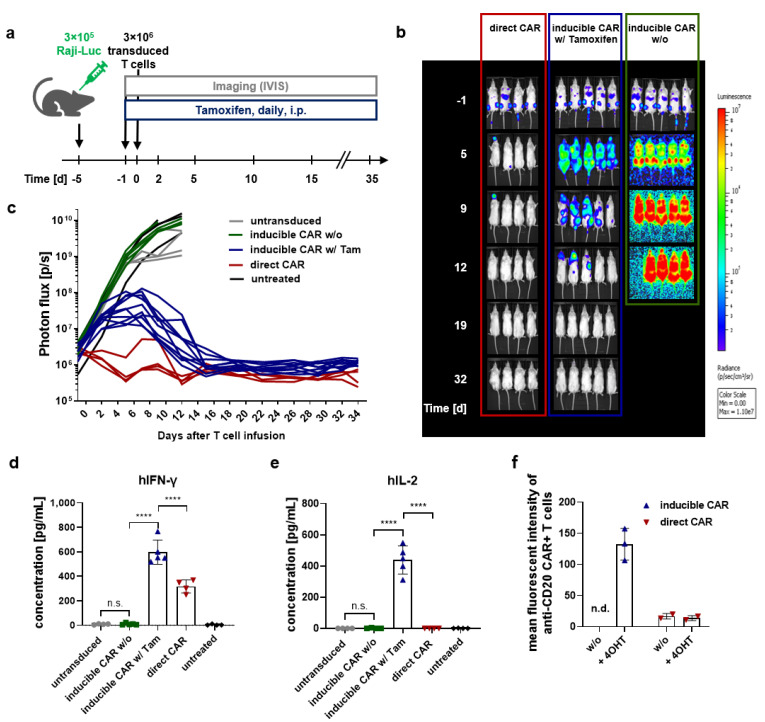
Cytotoxic activity of inducible anti-CD20 CAR T cells in vivo. (**a**) Schematic representation of the in vivo treatment schedule. NSG mice were inoculated with 3 × 10^5^ Raji^ffLuc^ cells via tail vein injection on day 5. On day 0, following randomization, mice were treated with untransduced, inducible, or conventional constitutively expressed anti-CD20 CAR T cells. Starting from day -1 tamoxifen was administered daily by i.p. injections to the untreated (= tumor only), untransduced, and inducible CAR w/ Tam groups, while the constitutively expressed direct CAR and the inducible CAR w/o groups received vehicle injections only. The tumor burden was regularly determined by in vivo BLI. (**b**) BLI images represent tumor progression in selected study groups. (**c**) Tumor burden expressed in photon flux (photons/sec) was determined and plotted over time for individual mice. (**d**,**e**) Human cytokines were measured in the plasma of individual mice 2 days after T cell infusion. Data represent means from *n* = 5 (inducible CAR ± Tam) and *n* = 4 (control groups). Data shown are mean values ± s.d. with **** *p* < 0.0001 by one-way analysis of variance (ANOVA). (**f**) Ex vivo cultivated T cells isolated via CD4^+^/CD8^+^ enrichment from the spleen of individual mice were induced with 100 nM 4-OHT for 46 h and analyzed for anti-CD20 CAR expression via flow cytometry. Data represent the MFI of anti-CD20 CAR. The frequency of anti-CD20 CAR expressing cells is shown in Appendix A.

## Data Availability

The datasets presented in this study are available from the corresponding author on reasonable request.

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
