# Peer review of "Titratable Pharmacological Regulation of CAR T Cells Using Zinc Finger-Based Transcription Factors"

_cancers, 2021, doi:10.3390/cancers13194741_

Round 1

Reviewer 1 Report

CAR T cell therapy has revolutionized treatment of refractory haematological malignancies. However, its side-effects can be lethal in a subset of patients. Effective safety mechanisms for CAR T cells could address this important clinical need.

In this manuscript, Kotter et al. show a novel control system for CAR expression in primary human T cells using a zinc-finger based transcriptional regulator for CARs. Most importantly, they illustrate both a negligent background-expression as well as a similar efficacy in vitro and in vivo compared to direct CAR expression. Furthermore, they show the possibility to fine-tune both the expression frequency and the surface presence (MFI) of the CARs. Its tunability potentially enables fine-tuning according to patient characteristics, and its strict transcriptional control could increase safety for CAR T cell treated patients. The authors should be commended for their important and innovative study. However, several caveats remain to be addressed before their findings can be interpreted completely.

Manuscript:

Line 142 - Contrary to what the authors state, MFI does not seem to "steadily increase over time" after 8 hours. Both frequency and MFI of CAR T cells (Fig. 1e) do not seem to significantly differ at 8h vs 22h.

Line 148 - Although stated as "significantly faster", it is not clear if frequency of GFP+ cells and MFI significantly differs for later time points after discontinuation of 4-OHT (Fig. 1h), or which statistical test was used. Futhermore, it remains unclear why 40h after 4-OHT addition only 20% of cells express GFP (compared to. 50% after 10h).

If access to equipment and reagents is no a hindrance, the authors could directly quantify CAR expression using qPCR, thus validating their stated (and central) hypothesis that the observed surface-persistence of the CAR after 4-OHT discontinuation only depends on the stability and half-life of the protein product.

Fig. 2g (right) suggests a decreased sensitivity to exhaustion for inducible CAR T cells compared to direct-CAR expressing T cells. This could be validated using FACS to asses presence of exhaustion markers in both settings.

As tamoxifen is highly lipophilic and the mouse model seems to produce very consistent results using the inducible anti-CD20 CAR T cells, it would be highly interesting to know if the effects seen in the intravenous Raji mouse model can be expanded to the subcutaneous Raji model, especially considering the increasing move of the CAR T cell field towards semi-solid (systemic & CNS lymphoma) and solid tumours.

Discussion:

A major limitation in the safety profile for tamoxifen-regulated transcription could be the relatively high half-life of tamoxifen (5-7 days) and its metabolites as well as the lingering presence of CARs on the surface of T cells after discontinuation even in vitro. This limits acute downregulation of the CAR at the surface level of T cells to times way beyond 1 week. This important limitation should be discussed.

Methods:

Time frame for experimental data should be stated (how many days after activation did the experiments start?) and kept consistent for both groups (inducible CARs vs direct CARs).

No reference and source is named for the destabilized GFP. Also, the vector for destabilized GFP expression is not shown. Does it use the same promoters? 

Figures:

Fig. 1b: legend seems to be missing

Fig. 1c: Full gating strategy should be disclosed (for example in supplementary information). Mean transduction efficiency (+/-S.D.) should be shown (% LNGFR+), is it different for the 2 different constructs? What does "normalized to the expression of the marker" mean, is it gated on LNGFR-positive cells (as in CAR+ cells divided by LNGFR+ cells and MFI of LNGFR+ cells), or is it (as stated in the figure) %LNGFR+ divided by %aCD20CAR+?

Fig. 2a: Do the E:T ratios illustrate total T cells, or is it normalized to transduced T cells and/or CAR+ T cells?

Fig. 2e: As it shows 2 donors, how can (at 10:1) the "inducible CAR w/o" have one measurement at 15%, and the mean at below 5%? The authors should consider setting "negative lysis" to zero. Also in Fig. S3c (0.1mg Tam - 24h).

Fig. 2f-g: As these show 2 donors, do these panels show pooled results or one representative of 2 seperately conducted experiments?

Fig. 4c, Fig. S3: Similar to panel 4b, Radiance (p/sec/cm2/sr) instead of mean photon flux (p/s) should be used.

Fig. 4f: Frequence of CAR+ cells could be included in addition to showing MFI.

Supplementary Information:

Ordering the panels and figures in the order they are mentioned in the manuscript could increase readability.

Fig. S2: What exactly is compared in 2b (**)? Direct CAR with inducible CAR w/o?

Fig. S3a: Could radiance be shown over time (similar to Fig. 4c) instead of only showing day 12?

Language:

Line 214 & 254 - on several occasions the authors refer to "tightness". Is it meant to show strict transcriptional control?

Reviewer 2 Report

The authors present data to support the feasibility of a zinc-finger based inducible system for transcriptional regulation of an anit-CD20 CAR T cell.  The investigators constructed a synthetic transcription factor composed of the N1 zinc finger protein, a modified ligand-binding domain of the estrogen receptor (G525R) and the transcriptional activation domain VP64.  In the presence of 4-OHT, the synthetic transcription factor binds to its respective binding sites and drives the transcription of the anti-CD20 CAR.  A constitutively expressed anti-CD20 CAR was used as a positive control.  Induction resulted in significantly higher levels of anti-CD20 CAR expression compared to control as early as four hours. Background levels of transgene expression were detected 10 days after discontinuation of  OHT treatment for inducible anti-CD20 CAR constructs. However, inducible CAR T cells showed efficient cytotoxic activity in the presence of 4-OHT, while CAR-related functions were lost after 4-OHT discontinuation in vitro.  In vivo, inducible CART cells with tamoxifen eradicated lymphoma within 19 days.  These mice remained in remission for at least 35 days of the study course.

This is a well written manuscript with interesting findings.

Can the authors comment on the potential impact of tamoxifen without a CAR on lymphoma response? 

Consider in the third paragraph of the discussion, briefly mention the side effect profile of tamoxifen.

Round 2

Reviewer 1 Report

The authors present an important addition to the existing control systems of CAR T cells.

The authors have thoroughly addressed all of my concerns and further improved their recommendable and highly relevant manuscript.